# Global Multi-Scale Optimization and Prediction Head Attentional Siamese Network for Aerial Tracking

Qiqi Chen [1,2] , Jinghong Liu [1,*], Xuan Wang [1], Yujia Zuo [1] and Chenglong Liu [1]

1 Changchun Institute of Optics, Fine Mechanics and Physics, Chinese Academy of Sciences, Changchun 130033, China; chenqiqi20@mails.ucas.ac.cn (Q.C.); wangxuan@ciomp.ac.cn (X.W.); zuoyujia@ciomp.ac.cn (Y.Z.); liuchenglong@ciomp.ac.cn (C.L.)
2 University of Chinese Academy of Sciences, Beijing 100049, China
* Correspondence: liujinghong@ciomp.ac.cn

**Abstract:** Siamese-based trackers have been widely used in object tracking. However, aerial remote tracking suffers from various challenges such as scale variation, viewpoint change, background clutter and occlusion, while most existing Siamese trackers are limited to single-scale and local features, making it difficult to achieve accurate aerial tracking. We propose the global multi-scale optimization and prediction head attentional Siamese network to solve this problem and improve aerial tracking performance. Firstly, a transformer-based multi-scale and global feature encoder (TMGFE) is proposed to obtain global multi-scale optimization of features. Then, the prediction head attentional module (PHAM) is proposed to add context information to the prediction head by adaptively adjusting the spatial position and channel contribution of the response map. Benefiting from these two components, the proposed tracker solves these challenges of aerial remote sensing tracking to some extent and improves tracking performance. Additionally, we conduct ablation experiments on aerial tracking benchmarks, including UAV123, UAV20L, UAV123@10fps and DTB70, to verify the effectiveness of the proposed network. The comparisons of our tracker with several state-of-the-art (SOTA) trackers are also conducted on four benchmarks to verify its superior performance. It runs at 40.8 fps on the GPU RTX3060ti.

**Keywords:** siamese-based tracking; aerial remote sensing tracking; multi-scale features; global context

## 1. Introduction

As a fundamental task in computer vision, visual object tracking mainly refers to the process of predicting the position and bounding box of an object in each subsequent frame of the video by using the information of the provided template in the initial frame. Aerial object tracking is widely used in many fields, such as video surveillance [1,2], automatic driving [3], aerial detection [4] and so on. However, in aerial remote sensing tracking, there are many challenges, such as scale variation, viewpoint change, occlusion, similar object and background clutter. These challenges make accurate and robust aerial tracking still a challenging task [5].

In order to handle these challenges, many researchers have invested a lot of energy in visual object tracking. DiMP [6] proposes an end-to-end structure that make full use of both template and background information to improve its discriminability. As a pioneering work based on the Siamese network, SiamFC [7] proposes a Siamese-based fully convolutional network that simplifies the calculation process of similarity prediction and shows the great potential of Siamese-based trackers. DaSiamRPN [8] modifies the sampling strategy of the training data to solve its distribution imbalance problem, which improves the tracking performance. AutoMatch [9] introduces six novel matching operators that replace the similarity calculation, which achieves tracking gains. SiamC-RPN [10] fuses multi-layer features and proposes a multi-RPN strategy to obtain more accurate bounding boxes. To

capture rich semantic correlation information, ACM-Siamese [11] proposes asymmetric convolution. Many other Siamese trackers [12,13] also adopt a Siamese-based network for visual tracking.

Although the above methods have improved the tracking accuracy to some extent, Siamese-based trackers are still limited to single-scale and local feature representation and are unable to meet the challenges of scale variation and occlusion during aerial remote sensing tracking. Additionally, for the prediction head, most trackers are limited to the receptive field of convolutional neural network and ignore the context information that is particularly important for dealing with similar object and background clutter.

Considering these two limitations of Siamese trackers, we propose two options to address these limitations and improve aerial tracking performance. One is replacing the single-scale features with multi-scale features and trying to add global information to improve its representation ability. In this way, we propose the transformer-based multi-scale and global feature encoder (TMGFE). During the feature extraction process, TMGFE first splits the feature map's channel into B groups and applies group convolution for multi-scale feature representation, then establishes multi-layer interactions through transformer to obtain global feature representation with semantics from different levels.

Another is designing an attention module to capture the response map's context information for the prediction head (PHAM). We know that the classification task aims to predict the label of each spatial location, and the regression task focuses on accurate target positioning. These two tasks perform different tracking tasks. Therefore, we do not directly add attention modules to the response map for context information but to both the classification branch and regression branch, respectively, for self-adaptive adjustment.

Some aerial remote sensing tracking results are displayed using three state-of-the-art (SOTA) trackers in Figure 1. We can see that challenges such as occlusion, viewpoint change and background clutter can deteriorate aerial tracking performance. Additionally, our tracker can handle these challenges and achieve more accurate tracking results.

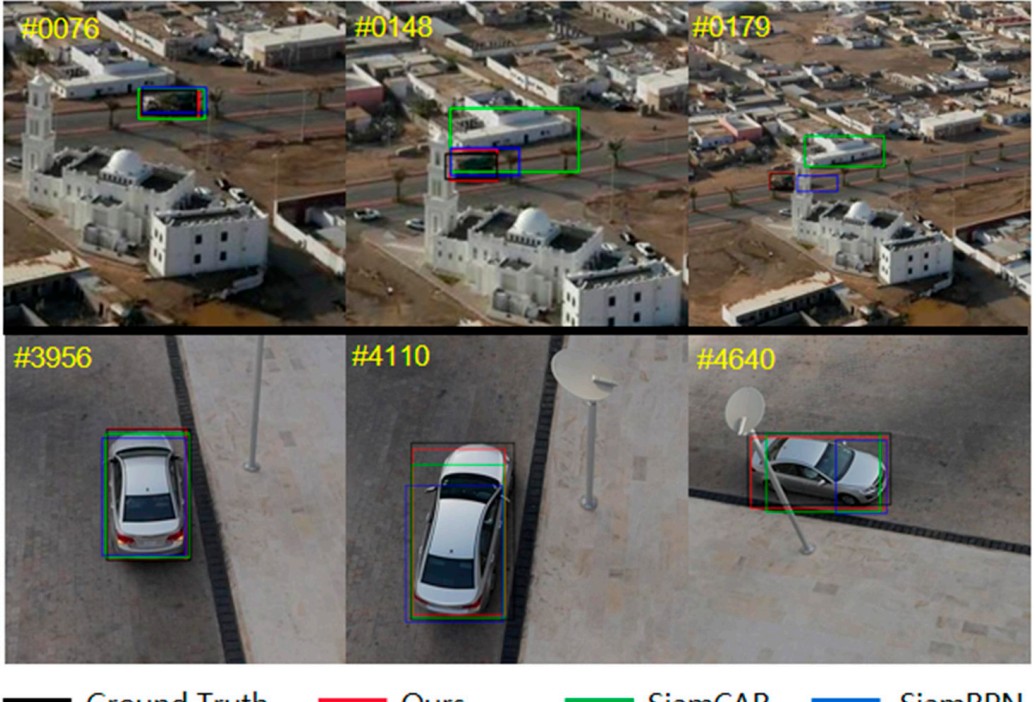

**Figure 1.** Examples of tracking performance under viewpoint change, occlusion and background clutter.

To sum up, we propose a Siamese network that extracts multi-scale global features and combines prediction head attention module to improve aerial remote sensing tracking performance. The contributions of this work are as follows:

1.  We propose a new Siamese network that extracts multi-scale global features and combines a prediction head attention module to solve challenges of aerial tracking and improve tracking performance.
2.  We have designed a new transformer-based multi-scale and global feature encoder (TMGFE). TMGFE first splits the channel dimension of each feature map and applies group convolutions for multi-scale spatial mixing, then establishes multi-layer interactions through the transformer to produce global feature representation with semantics from different levels.
3.  A prediction head attention module (PHAM) is proposed to adjust the spatial position contribution and the channel weight of the response map for capturing context information, thus reducing the impact of similar object and background clutter and achieving superior tracking performance.
4.  The performance of four benchmark datasets, including UAV123, UAV20L, UAV123@10fps and DTB70, proves the outstanding performance of the proposed network compared with several SOTA trackers.

## 2. Related Work

In this section, we first introduce Siamese-based trackers. Then, the multi-scale feature representation and transformer architecture are surveyed. Finally, the attention mechanism is briefly introduced.

### 2.1. Siamese-Based Trackers

In recent years, deep learning is used in many fields due to its powerful capability of feature extraction [14–16]. Siamese-based trackers also take advantage of deep learning and improve tracking performance, generating a great deal of interest in the visual tracking field.

A Siamese-based tracker is composed of two branches of neural network sharing weight. These two branches are used to extract features of object template and search area, calculate similarity scores between the template features and search candidates features, then return the most similar one as a tracking result by the learn to match function. SiamRPN [17] adopts a region proposal network that defines $k$ anchors with different sizes and proportions in advance for object scale estimation. SiamAPN [18] improves the anchor-based method by adaptively generating high-quality anchors for aerial tracking. SiamRPN++ [19] proposes a new spatial aware sampling strategy to break the restriction that Siamese-based trackers cannot use deep networks for feature extraction. However, the above Siamese trackers use an anchor-based mechanism for bounding boxes prediction, which makes the tracker sensitive to pre-defined parameters, such as the sizes and numbers of anchors, bringing difficulties to model training and poor generalization. The anchor-free Siamese-based tracker has received extensive attention. SiamCAR [20] adopts the anchor-free mechanism, which solves the problem of complicated parameter tuning and improves tracking accuracy. Also, SAM-DA [21] uses Siamese network for real-time nighttime aerial tracking. The Siamese network is widely used in aerial tracking due to its superior performance [22].

The above Siamese networks have improved tracking performance in different ways. However, they are limited to single-scale and local features and have poor tracking performance when dealing with scale variation, occlusion, similar object, background clutter and other challenges in aerial tracking. Additionally, due to the limitation of the receptive field, the convolutional neural network-based prediction head can only exploit contexts from local pixels and ignore the global information interactions, which may reduce aerial tracking accuracy.



Therefore, we designed a transformer-based multi-scale and global feature encoder to obtain multi-scale features with global context information; we also designed a prediction head attention module to improve the aerial tracking performance by explicitly extracting the context information for the prediction head.

### 2.2. Multi-Scale and Global Feature Encoder

Multi-scale representation has been used during feature extraction progress to improve the discrimination of network for a long time, such as the SIFT [23] feature.

GoogleNet [24] proposes the inception module which contains filters with different kernel sizes to improve the multi-scale representation of features. However, with the increase in kernel size, the amount of computation and parameters are greatly increased, which sacrifices the inference speed. Based on the inception module, CFPNet [25] applies dilation convolution to further improve the capability of extracting multi-scale information for real-time semantic segmentation. HRNet [26] obtains strong feature representations by performing different resolution convolution and information interaction. The feature representation of the results is not only powerful but also spatially accurate. SiamBAN [27] also obtains multi-scale features by establishing multi-layer connections.

Multi-scale features are widely used in computer vision tasks to improve performance. For the visual tracking field, the single-scale and local features struggle to meet the challenges of scale variation, viewpoint change, occlusion, background clutter and so on. Therefore, a transformer-based multi-scale and global feature encoder (TMGFE) is designed in this paper to obtain multi-scale features with global information to improve tracking accuracy.

### 2.3. Transformer

Transformer was first proposed in [28] and was applied in the field of Natural Language Processing to obtain global information from the input sequence. Later, ViT [29] and MobileViT [30] introduce transformer into computer vision, breaking the limitation that CNN can only obtain local information and ignore global information, establishing long-range connections. This strategy achieves superior results in object tracking. SparseTT [31] designs a sparse-attention-based transformer encoder and decoder to extract the most relevant information and achieve SOTA tracking performance. SiamTPN [32] designs a transformer-based feature pyramid module for a more robust target appearance. HiFT [33] proposes a hierarchical feature transformer that can extract both high-resolution and low-resolution features for aerial tracking.

Inspired by the development of the transformer and its ability to establish global connections, we designed a transformer-based multi-scale and global feature encoder to establish multi-layer and global interaction.

### 2.4. Attention Mechanism

In machine learning, the attention mechanism is used to measure the importance of input data to the output result [34]. The attention mechanism is introduced into the interpretable sentence embedding extraction model to improve the ability of natural language analysis. Subsequently, an attention mechanism is introduced into computer vision to help the module learn how to allocate its attention.

For example, SENet [35] proposes the "squeeze excitation module" that adaptively adjusts the importance of each channel by applying channel attention and enhances the model's representational capacity. CBAM [36] proposed an attention module that emphasizes important features from both channel and spatial dimensions, helping the network adaptively refine the feature and improve the performance of vision tasks.

In visual tracking, SiamAttn [37] designs a new attention mechanism to calculate self-attention and cross-attention. This mechanism enhances the information communication between the template branch and search branch and achieves implicit template update. SiamAPN++ [38] proposes an attention aggregation network that consists of self-attention

for self-interdependencies and cross-attention for cross-interdependencies, achieving real-time aerial tracking. This paper introduces a prediction head attention module. This module alleviates background interference and similar objects to some extent by increasing the useful channel and spatial information of the response map and suppressing the useless ones.

## 3. Methods

In this section, we mainly introduce the proposed aerial tracking network. First, we briefly introduce the architecture of the proposed Siamese network. Then, the transformer-based multi-scale and global feature encoder (TMGFE) and the prediction head attention module (PHAM) proposed in this paper are introduced in detail. Finally, we introduce the loss of anchor-free mechanism.

### 3.1. Network Architecture

As shown in Figure 2, there are four components in our network: feature extraction backbone network, transformer-based multi-scale and global feature encoder, cross-correlation module and anchor-free prediction head with attention-based attention module.

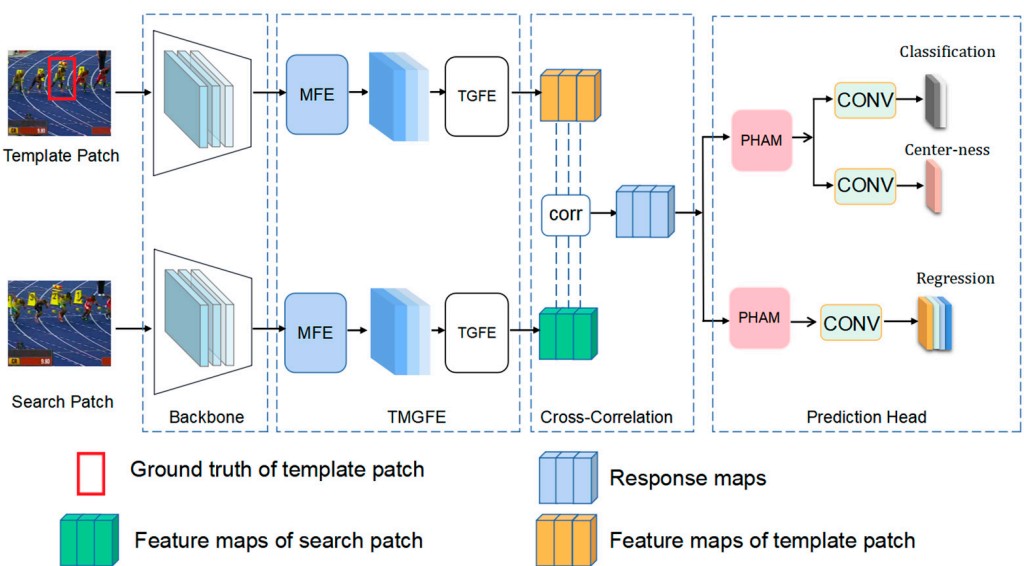

**Figure 2.** The architecture of our Siamese tracking network.

We first send the target template $Z$ and search area $X$ to the backbone network for feature extraction. After the backbone network, we obtain the feature maps $F_T$ of target template and $F_s$ of the search area. Then, we send the feature maps $F_T$ and $F_s$ to TMGFE, respectively. The TMGFE first splits the feature map's channels into B groups and applies $3 \times 3$ group convolution for multi-scale feature representation, then establishes multi-layer interactions through the transformer to obtain global feature representation with semantics from different levels.

The cross-correlation module adopts depth-wise cross-correlation to calculate the similarity between $F_T$ and $F_s$ and generates response maps. The response maps contain similarity information between the target template and the search area. Then, the prediction head attention module is used to capture context information of the response map, making the prediction head focus on more important information by adjusting the spatial position and channel contribution of the response map. Finally, the anchor-free classification network and regression network are used to obtain classification results and bounding boxes.

### 3.2. Transformer-Based Multi-Scale and Global Feature Encoder

For video sequences, effective trackers need to locate objects that may have scale variation and occlusion in different frames, while the backbone is mainly used to extract specific features for distinguishing a specific object category and cannot both make full use of global context information or estimate the scale variation of object well. Therefore, we propose a transformer-based multi-scale global feature encoder (TMGFE) to obtain multi-scale and global features for robust aerial tracking. There are two main components: multi-scale feature encoder (MFE) and transformer-based global feature encoder (TGFE).

Take the template branch as an example, the template patch T is sent to the backbone for feature extraction, and the backbone's last three blocks output three feature maps $L_1(Z), L_2(Z), L_3(Z) \in \mathbb{R}^{C \times H \times W}$. The Formula (1) represents that we compound $L_1(Z), L_2(Z), L_3(Z) \in \mathbb{R}^{C \times H \times W}$ as a unity $F_T \in \mathbb{R}^{3C \times H \times W}$ for subsequent processing.

$$F_T = \text{Concat}(L_1(Z), L_2(Z)L_3(Z)) \tag{1}$$

where Concat is the concatenation operation. $H$, $W$ and $C$ represent the size and the number of channels of each feature map of $F_T$.

We take $L_1$ as an example, MFE splits its channels into $k$ subsets. We denote each subset as $x_i \in \mathbb{R}^{C/k \times H \times W}$, where $i$ belongs to $\{1, 2, \ldots, k\}$. We send every subset to $3 \times 3$ depth-wise convolution (represented as $d_i$) and the output is $y_i$; $y_i$ is added to the next subset $x_i + 1$ and fed into $d_{i+1}$. After that, we concat every $y_i$, where $i$ belongs to $\{1, 2, \ldots, k\}$. Through these different filter groups of $3 \times 3$ depth-wise convolution, we obtain the multi-scale feature map $L_1'$. Each group convolution and residual connection between subsets also increases the network's receptive field. The specific formula is as follows:

$$y_i = \begin{cases} d_i(x_i) & i = 1 \\ d_i(x_i + y_{i-1}) & 2 \leq i \leq k \end{cases} \tag{2}$$

$$L_1' = \text{Concat}(y_i), i \in \{1, 2, \ldots, k\} \tag{3}$$

If we do the same operation for both $L_2$ and $L_3$, we obtain $L_2'$ and $L_3'$. Finally, we obtain multi-scale feature maps $F_T'$.

$$F_T' = \text{Concat}(L_1', L_2', L_3') \tag{4}$$

Our encoder not only uses channel splitting and group convolution but also attempts to communicate multi-layer information to enhance the representation of multi-scale features. Inspired by the ability to model the global dependencies of transformer, we design a transformer encoder to establish information interaction between multiple layers for global information and further enhance the representation of multi-scale features. Compared with a single convolution operation, multi-scale features can enhance the discriminative capacity of this network.

As shown in Figure 3, Our encoder contains the attention module, feed-forward network and layer-normalization. To establish information interaction between multiple layers ($L_1'$, $L_2'$ and $L_3'$), the intermediate feature layer $L_2'$ is used as the query and the feature map ($L_1'$ or $L_3'$) itself as the key and value. The intermediate feature layer $L_2'$ contains more balanced semantic and spatial information that can be used as query $Q$, and $L_1'$ and $L_3'$ can be used as the key $K$ and value $V$ for global interaction.

$$\text{Attention}(Q, K, V) = \text{Softmax}(\frac{Q \cdot K^T}{\sqrt{d_k}}) \cdot V \tag{5}$$

where $d_q = d_k = d_v = 256$ means the number of channels of the feature map. Specifically, for establishing information interaction between $L_1'$ and $L_2'$, we set $Q = L_2' \in \mathbb{R}^{C \times H \times W}$ and $K = V = L_1' \in \mathbb{R}^{C \times H \times W}$. The $Q \cdot K^T$ is exploited to calculate the degree of attention between

two different feature maps; $\sqrt{d_k}$ is used for normalization calculation and "Softmax($\cdot$)" is utilized as the activation function to obtain the weight value for $V$.

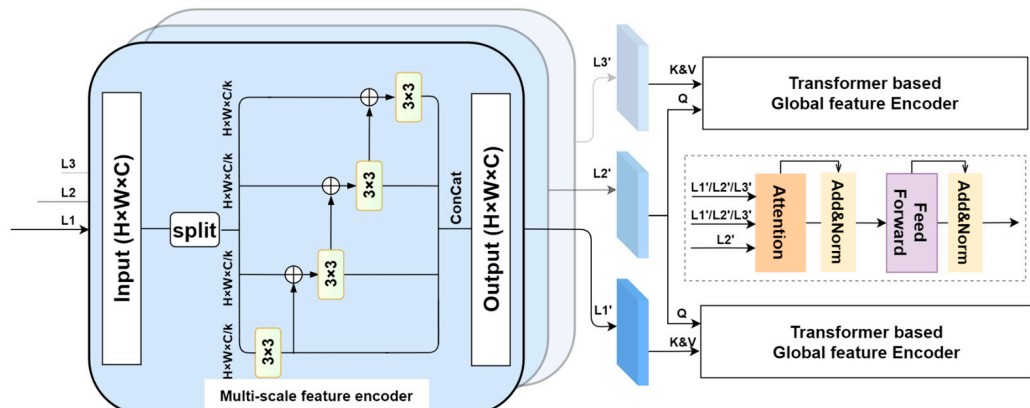

**Figure 3.** The structure and flowchart of our transformer-based multi-scale and global feature encoder.

The output of "Attention($\cdot$)" is added to input $V$ as the residual connection, then we send the result to layer-normalization for normalization calculation. The formula can be expressed as follows:

$$y_z = \text{LayerNorm}(V + \text{Attention}(Q, K, V)) \tag{6}$$

A fully connected network with ReLu activation function is applied as a feed-forward network, followed by residual connection and layer-normalization. The formulas are as follows:

$$y_z' = \text{LayerNorm}(y_z + \text{FFT}(y_z)) \tag{7}$$

$$\text{FFT}(y_z) = W_2 \cdot \text{ReLu}(y_z + b_1) + b_2 \tag{8}$$

where ReLu means the activation function, $W_i$ and $b_i$ ($I$ = 1, 2) represent parameter weight and biases. Taking $L_1'$ as an example, we use $L_2'$ as query, $L_1'$ as the key and value, then we use formulation (5) to calculate the global information interrelation between $L_2'$ and $L_1'$. The result is added to $L_1'$ as residual connection, then we apply layer-normalization and feed-forward and layer-normalization again to establish information interaction between $L_2'$ and $L_1'$.

In conclusion, TMGFE is proposed to obtain multi-scale and global features that can produce robust target-specific appearance representation and improve aerial remote sensing tracking performance.

### 3.3. Prediction Head Attention Module

After the TMGFE, we obtain the multi-layer and global feature maps of the target template $F_T'$ and the search area $F_S'$. By calculating the cross-correlation between $F_T'$ and $F_S'$, where $F_T'$ is used as the kernel for cross-correlation to $F_S'$, we obtain the similarity maps. The similarity maps $R$ establish the connection between the template branch and the search branch. The formula can be expressed as follows:

$$R = F_T' \otimes F_S' \tag{9}$$

where "$\otimes$" means the cross-correlation.

The prediction head consists of two subtasks: the classification task and the regression task. It is easy to understand that through decoding the response map, we can obtain the bounding box of the search area that is most closely related to the object template as the prediction results. However, in aerial tracking, the background clutter will pollute

the features of the target and similar objects usually have the same semantic information as the target. These challenges will interfere with the response map, e.g., the response map may produce a high response besides the true target, which may increase the risk of tracking drift. Therefore, we propose the prediction head attention module to add context information and reduce the impact of these challenges.

As shown in Figure 4, we adopt the prediction head attention module (PHAM) to explore the context information of the response map in two dimensions: channel and spatial location. Channel attention can help prediction focus on more important features, while spatial attention considers more spatial information that can improve the results of bounding boxes.

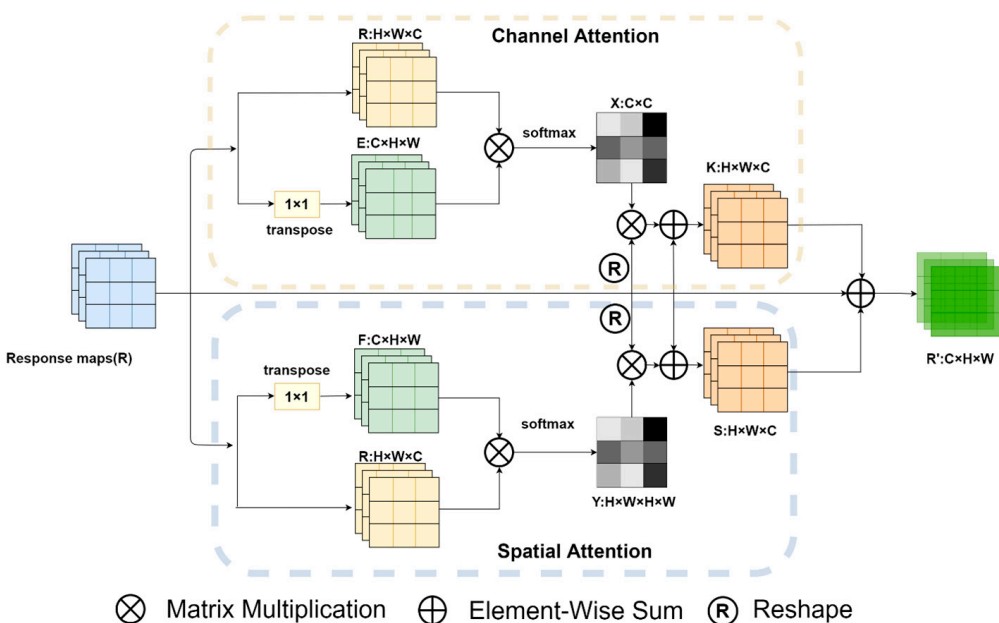

**Figure 4.** The architecture of the proposed prediction head attention module.

As a result, our PHAM improves tracking performance by increasing the useful channel and spatial information and suppressing the useless ones. In Figure 5, we visualized the classification with or without PHAM; we can clearly see that, under similar object and background clutter, the response map generates multiple responses that will cause interference to the prediction head and increase the risk of the network being fooled by distractions. We can also see that the classification map with the proposed PHAM can reduce the impacts of these challenges, which will help the prediction head recognize the target from its background and achieve better performance in aerial remote sensing tracking.

For the channel attention module in particular, each channel in the response map is updated by combining its features with a weighted summation, where the weights are based on how each channel impacts the others. We first use the $1 \times 1$ convolution to transpose the response map from $R \in \mathbb{R}^{H \times W \times C}$ to $E \in \mathbb{R}^{C \times H \times W}$ to calculate the coefficient matrix $X$ of all channels using the following formula:

$$X \in R^{C \times C} : X_{ji} = \frac{\exp(E_i \cdot R_j)}{\sum\limits_{i=1}^{C} \exp(E_i \cdot R_j)} \tag{10}$$

where $X_{ji}$ is the impacts of $E_i$ (the $i$th channel of $E$) on $R_j$ (the $j$th channel of $R$).

$$K \in R^{C \times H \times W} : K_j = \alpha \cdot \sum\limits_{i=1}^{N} \exp(X_{ji} \cdot R_j) + R_j \tag{11}$$

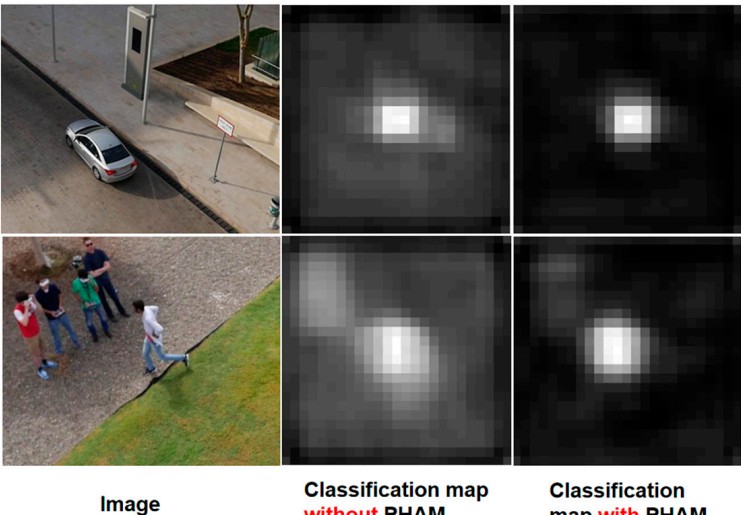

| Image | Classification map without PHAM | Classification map with PHAM |

**Figure 5.** Visualization of Classification map. From left to right: search image, Classification map without PHAM, Classification map with PHAM.

Multiply the obtained $X_{ji}$ with $R_j$ to readjust the channel weight of the response map, then multiply it by a scale parameter $\alpha$, which is gradually determined during training. And the output is connected to the original response map to enhance the response map with context information.

For the spatial attention module, every spatial position of the response map is updated by combining the general characteristics of all positions with a weighted summation, where the weights are based on the correlation between the two positions. We perform a similar operation on the spatial attention module, and the formula is as follows:

$$Y \in R^{N \times N} : X_{ji} = \frac{\exp(F_i \cdot R_j)}{\sum\limits_{i=1}^{N} \exp(F_i \cdot R_j)}, N = H \times W \tag{12}$$

where $Y_{ji}$ is the impacts of $F_i$ (the spatial position of $i$th) on $R_j$ (the $j$th the spatial position of $R$).

$$S \in R^{C \times H \times W} : S_j = \beta \cdot \sum_{i=1}^{N} \exp(Y_{ji} \cdot R_j) + R_j \tag{13}$$

where $\beta$ is the scale parameter that is gradually determined during training.

$$R' = S + K + R \tag{14}$$

where $S$ is the feature map after spatial attention enhancement, and $K$ is the feature map after channel attention enhancement. $R$ is the original feature map. Then, we add $S$ and $K$ to $R$, in this way, we obtain the adjusted response map $R'$.

To sum up, after capturing the response map, both channel and spatial attention are used for feature enhancement, and the global context information is introduced to reduce tracking failure. As mentioned above, the prediction head attention module proposed in this paper adaptively adjusts the spatial position and channel weight to achieve more accurate classification and regression prediction. Global context information is introduced to reduce tracking failures. Finally, we chose an anchor-free mechanism to simplify the training progress and reduce the influence of prior knowledge on the generalization of our tracker.

### 3.4. Training Loss

For anchor-free trackers, the prediction head outputs the classification map $A_{cls} \in \mathbb{R}^{H \times W \times 2}$ for the classification task and the regression map $A_{reg} \in \mathbb{R}^{H \times W \times 4}$ for the regression task.

For the classification feature map $A_{cls} \in \mathbb{R}^{H \times W \times 2}$, the H and W represent the size of the feature map and each point consists of both target score $\delta_{pos}$ and background score $\delta_{neg}$ calculated by the classification branch. For the regression feature map $A_{reg} \in \mathbb{R}^{H \times W \times 4}$, each point $(i, j, :)$ contains a 4D vector $(l, t, r, b)$ that represents the distances of left, top, right and bottom four sides to the corresponding position and is recorded as $t(i, j) = (l, t, r, b)$. The regression bounding box $T_{(i,j)}$ at search patch $(x, y)$ can be calculated using Formula (15).

$$
\begin{aligned}
l' = x - x_0, t' = y - y_0 \\
r' = x_1 - x, b' = y_1 - y
\end{aligned}
\tag{15}
$$

where the $(x_0, y_0)$ is the left-top corner of ground truth and $(x_1, y_1)$ represents the right-bottom corner. We use the regression bounding box $T_{(i,j)}$ and the predicted bounding box $t_{(i,j)}$ to compute the regression loss $L_{reg}$ :

$$
L_{reg} = \frac{1}{\sum\limits_{(i,j)} \prod(T_{(i,j)})} \sum_{(i,j)} \prod(T_{(i,j)}) \times L_{IoU}\left[T_{(i,j)}, t_{(i,j)}\right]
\tag{16}
$$

where $L_{IoU}[T_{(i,j)}, t_{(i,j)}]$ is the *IoU* loss of $T_{(i,j)}$, and $\mathbb{I}(i, j)$ is an indicator function:

$$
\prod(i, j) = \begin{cases} 1 & l', t', r', b' > 0 \\ 0 & \text{otherwise} \end{cases}
\tag{17}
$$

where $l', t', r', b' > 0$ means that the point $(i, j)$ exists within the ground truth. For the classification loss, widely used BCELoss in the classification task is our choice, and the specific formula is as follows:

$$
L_{cls} = 0.5 \times \text{BCELoss}(\delta_{pos}, I) + 0.5 \times \text{BCELoss}(\delta_{neg}, I)
\tag{18}
$$

where $I$ is the classification grounding truth of the position. We also introduce a center-ness branch to remove outliers and generate the center-ness feature map $A_{cen} \in \mathbb{R}^{H \times W \times 1}$ that contains the center-ness score $C(i, j)$ of each location. Use the formula to calculate the center-ness score when the sample point is in the foreground.

$$
C(i, j) = I \times \sqrt{\frac{\min(l, t)}{\max(l, t)} \times \frac{\min(r, b)}{\max(r, b)}}
\tag{19}
$$

We can see from Formula (19) that $C(i, j)$ is related to the distance between the target center and the corresponding location $(x, y)$. $C(i, j)$ obtains a max value of 1 if $(x, y)$ is exactly the target center, otherwise set $C(i, j)$ as 0. And the center-ness loss is

$$
L_{cen} = \frac{1}{\sum \prod(i, j)} \sum_{(i,j)} L_{IoU}(A_{cen}(i, j, :), T_{(i,j)}) + (1 - C(i, j)) \log(1 - A_{cen}(i, j, :))
\tag{20}
$$

The overall loss function is

$$
L = \alpha_1 L_{cls} + \alpha_2 L_{cen} + \alpha_3 L_{reg}
\tag{21}
$$

The total loss consists of the classification loss $L_{cls}$, the center-ness loss $L_{cen}$ and the regression loss $L_{reg}$, where $\alpha_1$, $\alpha_2$ and $\alpha_3$ are used as the weight of $L_{cls}$, $L_{cen}$ and $L_{reg}$.

## 4. Results

In this section, the training and testing progress is first illustrated in detail, then the test benchmarks and metrics are introduced. The ablation experiments are performed to verify the effectiveness of the proposed tracker for aerial remote sensing tracking. Finally, we illustrate the performance of the proposed network by presenting the tracking results and comparison with some SOTA trackers in UAV123 [39], UAV20L [39], UAV123@10fps [39] and DTB70 [40].

### 4.1. Training and Testing Detail

We train our network by using four datasets, including GOT-10K [41], COCO [42], LaSOT [43] and VID [44]. The RTX3060ti was chosen for training in Python 3.7 and Pytorch 1.7.1 versions. We chose ResNet50, pre-trained on ImageNet, as the backbone network for feature extraction. Then, we fine-tuned our tracker with stochastic gradient descent and set the batch size to 12. The total training epoch was 20 for the first 5 epochs, and we used 0.001 to 0.005 as the warm up learning rate. For the next 15 epochs, the learning rate was set at 0.005 to 0.0005.

### 4.2. Evaluation Benchmarks
#### 4.2.1. UAV123 Benchmark

The UAV123 benchmark consists of 123 video sequences that are both captured from aerial perspectives. UAV123 benchmark contains normal remote sensing scenes like bake, boat, car, group, person, truck, uav and wakeboard. Additionally, UAV123 contains 11 challenges of aerial tracking, such as occlusion, illumination variation, large-scale variations, fast motion, similar object and background clutter. The precision rate and success rate are used to evaluate the tracking result on the UAV123 benchmark. The distance between the bounding box's center and the ground truth is measured as precision. Success measures the Intersection over Union (IoU) score between the predicted result and the ground truth bounding box.

#### 4.2.2. UAV20L Benchmark

The UAV20L benchmark is an aviation video dataset for long-term tracking that contains 20 long-term sequences. These sequences are both captured from aerial perspectives. The average number of frames in UAV20L benchmark is 2934 and the maximum sequence contains 5527 frames. UAV20L also contains remote sensing scenes like bike, bird, car, group, person and uav and is widely used as a benchmark for long-term aerial tracking. UAV20L uses precision and success rate to evaluate the tracking results.

#### 4.2.3. UAV123@10fps Benchmark

UAV123@10fps benchmark contains 123 sequences whose frame rates are 10 fps. This benchmark captures cars, groups, boats and so on from aerial perspectives. This benchmark contains 12 challenges of aerial tracking, such as occlusion, scale variations, fast motion, similar object and background clutter. Therefore, UAV123@10fps is appropriate to evaluate trackers. UAV123@10fps benchmark uses precision and success rate to evaluate the tracking results.

#### 4.2.4. DTB70 Benchmark

DTB70 benchmark contains 70 video sequences that are all captured from aerial perspectives. DTB70 benchmark contains remote sensing scenes like Gull, RcCar and Yacht and is widely used as a benchmark for aerial tracking. DTB70 uses precision and success rate to evaluate the tracking results.

### 4.3. Ablation Experiments

To prove the effectiveness of the proposed TMGFE and PHAM, we conducted ablation experiments on the UAV123 benchmark and the UAV20L benchmark. The first experiment

was to add the transformer-based multi-scale and global feature encoder (TMGFE) to a Siamese tracker and compare the tracking results with or without TMGFE. The second experiment was to verify the prediction head attention module (PHAM) proposed in this paper. We add PHAM to a Siamese tracker and compare the tracking results. The third experiment was to add both TMGFE and PHAM to the Siamese tracker and compare the tracking results.

It is not difficult to see from Table 1 that the success rate and the precision rate of the Siamese tracker with TMGFE attain 0.810 and 0.620, 1.0% and 2.0% higher than the tracker without TMGFE on the UAV123 benchmark. The precision rate and the success rate achieve 0.733 and 0.562, respectively, 2.7% and 3% higher than the tracker without TMGFE on the UAV20L benchmark. Also, we can see the results of the Siamese tracker with PHAM attain 0.815 and 0.619, 1.5% and 1.9% higher than the tracker without PHAM on the UAV123 benchmark. The precision rate and the success rate achieve 0.731 and 0.563, 2.5% and 3.1% higher on the UAV20L benchmark.

**Table 1.** Ablation study of TMGFE on the UAV123 and UAV20L.

| NO | ResNet-50 | Corr | Head | TMGFE | PHAM | UAV123 | | UAV20L | | Fps |
|----|-----------|------|------|-------|------|--------|------|--------|------|-----|
| | | | | | | Pre (%) | Succ (%) | Pre (%) | Succ (%) | |
| 1 | √ | √ | √ | | | 80.0 | 60.0 | 70.6 | 53.2 | 42 |
| 2 | √ | √ | √ | √ | | 81.0 | 62.0 | 73.3 | 56.2 | 41.3 |
| 3 | √ | √ | √ | | √ | 81.5 | 61.9 | 73.1 | 56.3 | 41.7 |
| 4 | √ | √ | √ | √ | √ | 82.7 | 63.6 | 76.1 | 58.1 | 40.8 |

Moreover, the success and precision of the Siamese tracker with both TMGFE and PHAM attain 0.827 and 0.636, 2.7% and 3.6% higher than the base Siamese tracker on UAV123 benchmark. The contributions of the precision rate and the success rate of 0.761 and 0.581 also improved by 5.5% and 4.9% on the UAV20L benchmark. Additionally, the fps of our tracker is 40.8 and meets the requirement for real-time tracking.

The results in Table 1 prove that the proposed TMGFE and PHAM can significantly improve tracking performance, achieve precision and success gain. To further demonstrate the effectiveness of TMGFE, we visualized the heat maps of the Siamese tracker with or without TMGFE.

We can see from Figure 6a that the Siamese tracker equipped with TMGFE can focus on the target and identify the boundary of the object better when meeting the challenges of scale variation and viewpoint change, which proves that TMGFE can effectively extract multi-scale features and greatly improve the performance of tracking. Figure 6b also shows that the tracker equipped with PHAM can reduce the interference of clutter and achieve accurate tracking while the results without PHAM fail, which also proves the effectiveness of the proposed PHAM.

In summary, through experimental data comparison and heat map analysis, we prove the effectiveness of the proposed TMGFE and PHAM in aerial tracking and obtain the best tracking result when the Siamese network is equipped with both TMGFE and PHAM.

### 4.4. State-Of-The-Art (SOTA) Comparisons

4.4.1. Tracking Comparison on UAV123

Figure 7, shows the comparisons between our tracker and nine SOTA trackers, including Ocean [45], SiamRPN++ [19], SiamTPN [32], SiamBAN [27], HiFT [33], SiamFC++ [46], SiamDW [47], SiamSA [48] and SiamCAR [20].

As shown in Figure 7, Our network ranks first among all comparison trackers, the success rate is 0.639, 0.3% higher than SiamTPN, and the precision score is 0.832, 0.9% higher than Ocean. In addition, the radar chart analysis was also conducted for multiple challenge attributes of the UAV123 benchmark. As shown in Figure 8, our tracker performed better than the comparison algorithm in nine challenges of UAV, including scale variation, partial

occlusion and so on. The comparisons prove that our network can solve the challenges in aerial tracking to some extent.

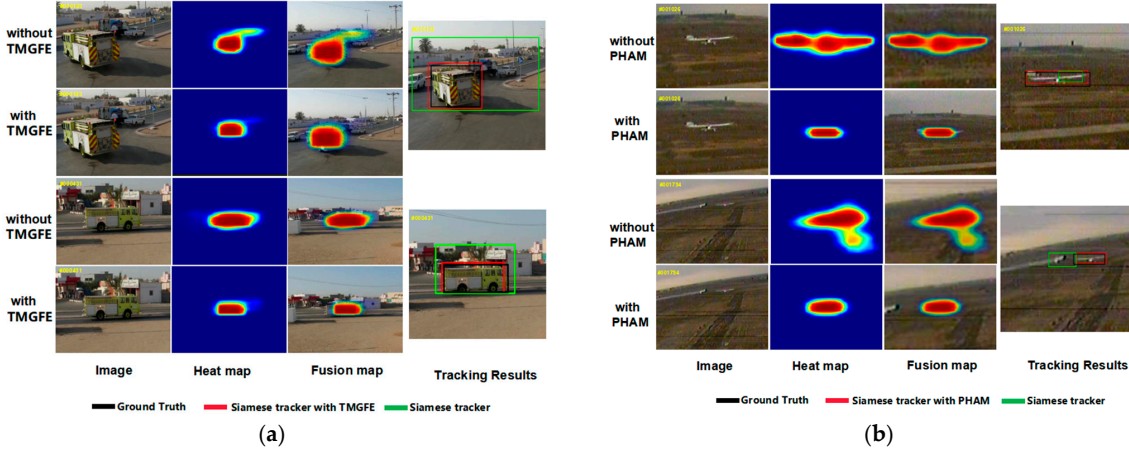

**Figure 6.** Visualization of heat map. (**a**) From left to right: search image, heat maps of features with or without TMGFE, fusion maps of features with or without TMGFE, the tracking results with or without TMGFE. (**b**) From left to right: search image, heat maps of features with or without PHAM Encoder, fusion maps of features with or without PHAM Encoder, the tracking results with or without PHAM Encoder.

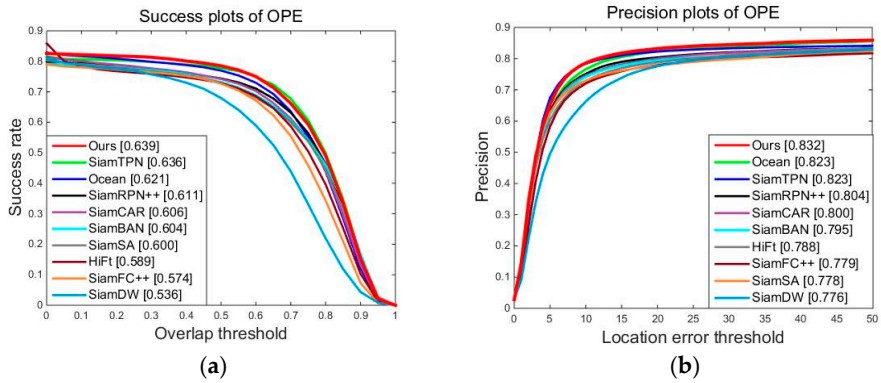

**Figure 7.** Success plots (**a**) and precision plots (**b**) on UAV123 benchmark.

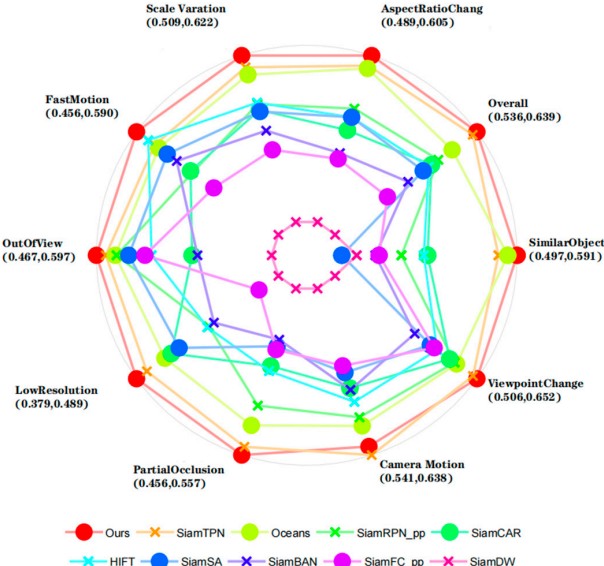

**Figure 8.** The success rate on UAV123 for nine challenging attributes.

#### 4.4.2. Tracking Results on UAV20L

The comparisons between our tracker and six SOTA trackers ECO [49], SiamFC [7], SiamFC++ [46], SiamAPN [18], SiamAPN++ [38], SiamRPN [19] and DaSiamRPN [8] are shown in Table 2.

**Table 2.** UAV20L benchmark comparison table.

| Tracker | ECO | SiamFC | DaSiamRPN | SiamFC++ | SiamAPN | SiamAPN++ | Ours |
|---------|-----|--------|-----------|----------|---------|-----------|------|
| Succ (%) | 42.7 | 40.2 | 46.5 | 53.3 | 54.0 | 56.1 | 58.1 |
| Pre (%) | 58.9 | 59.9 | 66.5 | 69.5 | 72.0 | 73.6 | 76.1 |

We can see from Table 2 that the success score of our tracker is 0.581, 2% higher than SiamAPN++, and the precision score is 0.761, 2.5% higher than SiamAPN++. Compared with these SOTA trackers, we obtain the best results on the UAV20L benchmark, which proves that the proposed network can not only deal with the challenging scenarios of aerial tracking but also achieve long-term aerial tracking.

#### 4.4.3. Tracking Comparison on UAV123@10fps

Figure 9 shows the comparisons of our tracker with nine SOTA trackers on UAV123@10fps benchmark, including TCTrack++ [50], SiamCAR [20], SiamAPN [18], SGDViT [51], SiamSA [48], SiamAPN++ [38], SiamBAN [27], HiFt [33] and AutoTrack [52]. Success plots and precision plots are shown in Figure 9, where our tracker achieves the best tracking result; the success rate and the precision are 63.7% and 80.5%. Compared with TCTrack, we obtain a 3.7% improvement in success rate and a 2.5% improvement in precision rate. We can see from Figure 10, the success rates under background clutter, scale variation, deformation and the other six challenges prove that the proposed TMGFE and PHAM can improve the tracking performance in these challenging scenes.

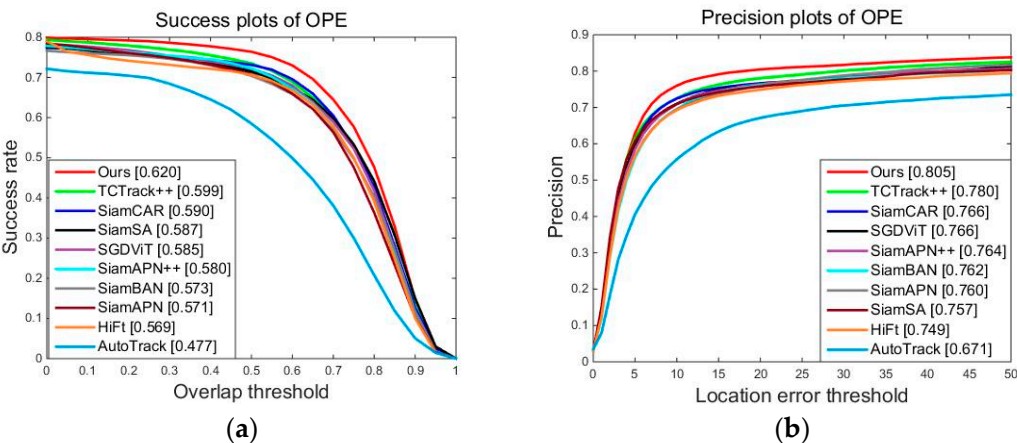

**Figure 9.** Success plots (**a**) and precision plots (**b**) on UAV123@10fps.

#### 4.4.4. Tracking Results on DTB70

We also compare our algorithm with seven SOTA trackers, including Ocean [45], SE-SiamFC [52,53], SiamAPN++ [38], HiFT [33], SGDViT [51], SiamCAR [20], and SiamAttn [37] on DTB70 benchmark to validate the tracking performance of the proposed tracker. The experimental results are shown in Table 3. The metrics of precision, Norm precision and success of our tracker achieved 84.4%, 80.7% and 65.6%, 1.6%, 1.8% and 1.1% higher than SiamAttn, obtaining the best results of all comparison trackers.

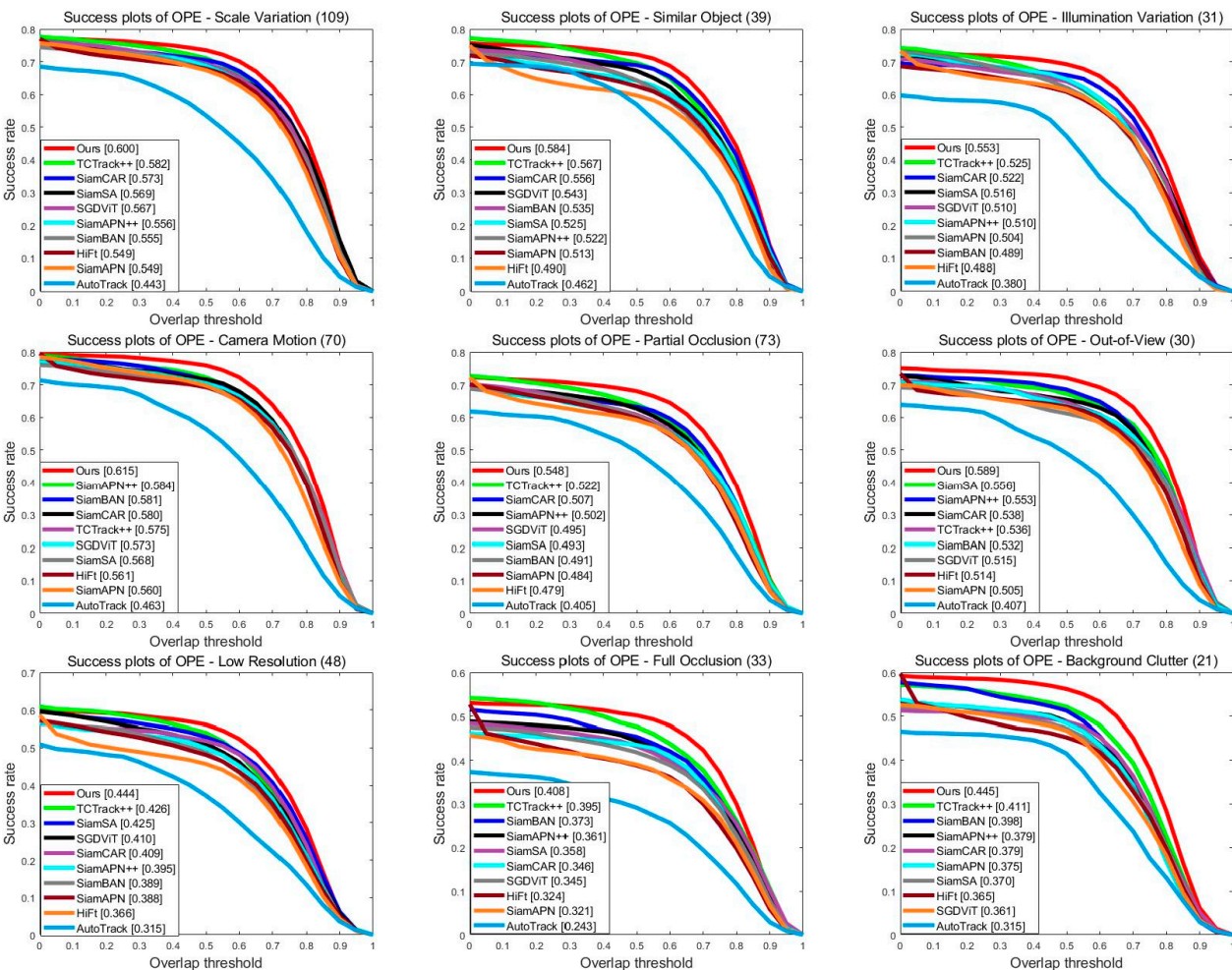

**Figure 10.** The success comparison with nine SOTA tracker of 12 attributes on UAV123@10fps.

**Table 3.** Comparisons on DTB70.

| Tracker | Pre (%) | Norm (%) | Succ (%) |
|---------|---------|----------|----------|
| Ocean | 63.4 | 53.3 | 45.5 |
| SE-SiamFC | 73.0 | 57.3 | 49.6 |
| SiamAPN++ | 79.0 | 65.5 | 59.4 |
| HiFT | 82.0 | 65.5 | 59.4 |
| SGDViT | 86.0 | 67.3 | 63.0 |
| SiamCAR | 83.9 | 79.1 | 64.5 |
| SiamAttn | 82.8 | 78.9 | 64.5 |
| Ours | 84.4 | 80.7 | 65.6 |

*4.5. Qualitative Analysis*

To further demonstrate the performance of our tracker in aerial tracking, we performed a tracking visualization comparison of our tracker with three SOTA trackers, namely SiamRPN++, SiamCAR and SiamFC, on four challenging video sequences on the UAV123 benchmark. In addition, we have also compared our tracker visually with three SOTA trackers, SiamCAR, SiamAttn and SGDViT, on the DTB70 benchmark. The tracking results are shown in Figure 11. In the Car9_1 sequence, our tracker is able to track successfully when challenged by occlusions and similar object clutter. Among all the compared trackers, only our tracker was successful in tracking when the target undergoes dramatic occlusion, which proves the effectiveness of the multi-scale and global features extracted by our tracker; in the Car7_1 sequence, there is a challenging scenario containing occlusion, similar

objects and background clutter. As can be seen in Figure 11, our tracker can cope with these challenges, while SiamFC and SiamRPN++ show tracking drift and the IoU scores of SiamCAR are not as high as ours, which proves that our proposed network can minimize the impact of these challenges; In the Bike2_1 sequence, only our tracker achieves accurate tracking, while SiamRPN++, SiamCAR and SiamFC fail due to partial occlusion and background clutter. In the Boat3 sequence, there is scale variation, viewpoint variation and background clutter. In all comparison trackers, our tracker achieves accurate aerial tracking and fits the ground truth well, which demonstrates the effectiveness of the proposed tracker. In the Gull2 sequence, the target undergoes scale changes and background clutter; as can be seen in Figure 12, our tracker can handle the background clutter and achieve accurate tracking throughout, whereas SiamCAR, SiamAttn, and SGDViT all lose the target and fail to track. In the RcCar sequence, there are challenges of background clutter and low resolution, as can be seen in Figure 12, and our tracker consistently tracks the target. In the Yacht4_1 sequence, the object experienced severe occlusion and scale variations, and only our tracker was able to track the target accurately at all times.

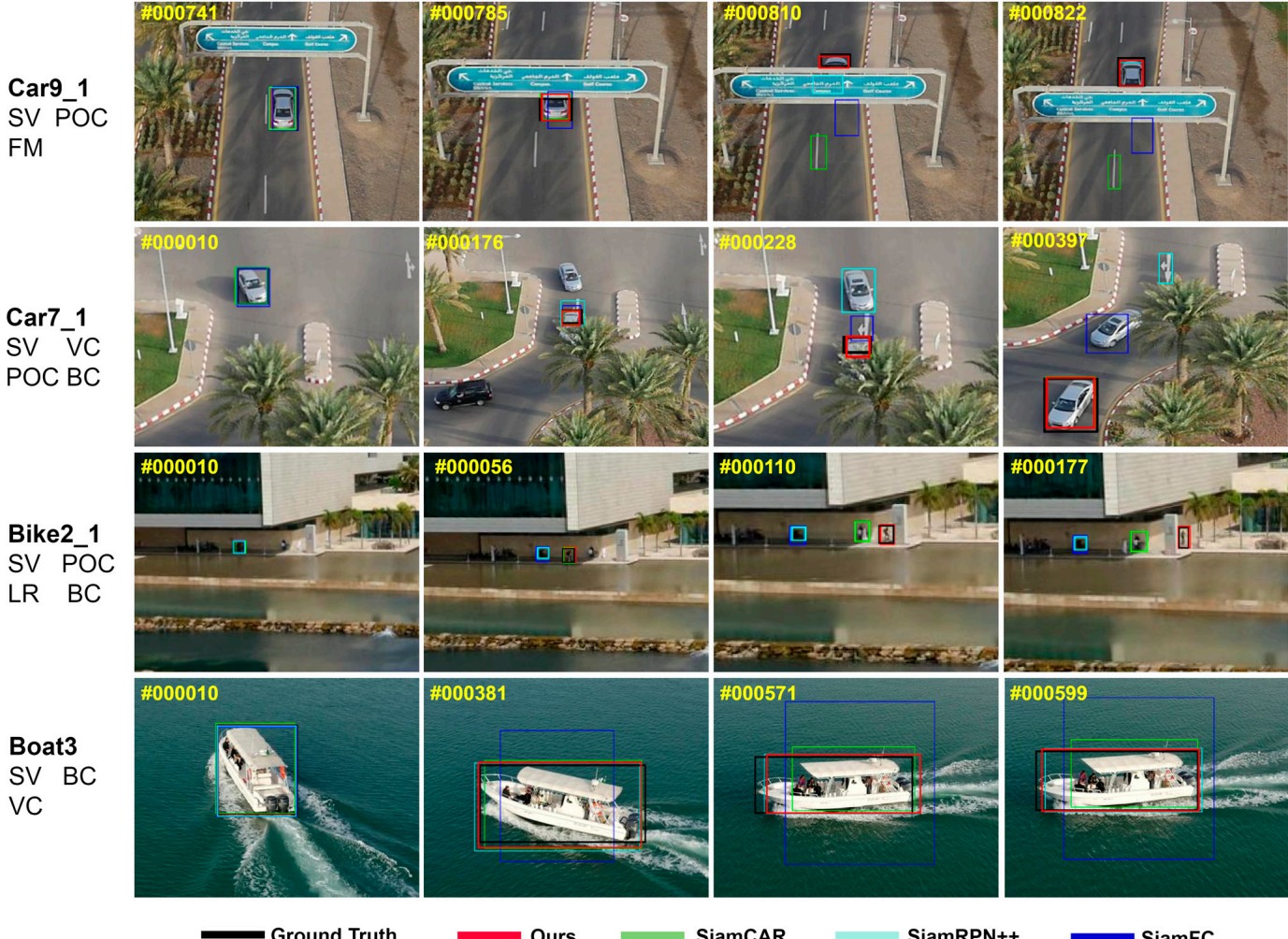

**Figure 11.** Qualitative analysis of our network and three SOTA trackers on car9_1, car7_1, truck4, car15, boat3 and bike2_1 aerial remote sensing sequences in UAV123 benchmark.

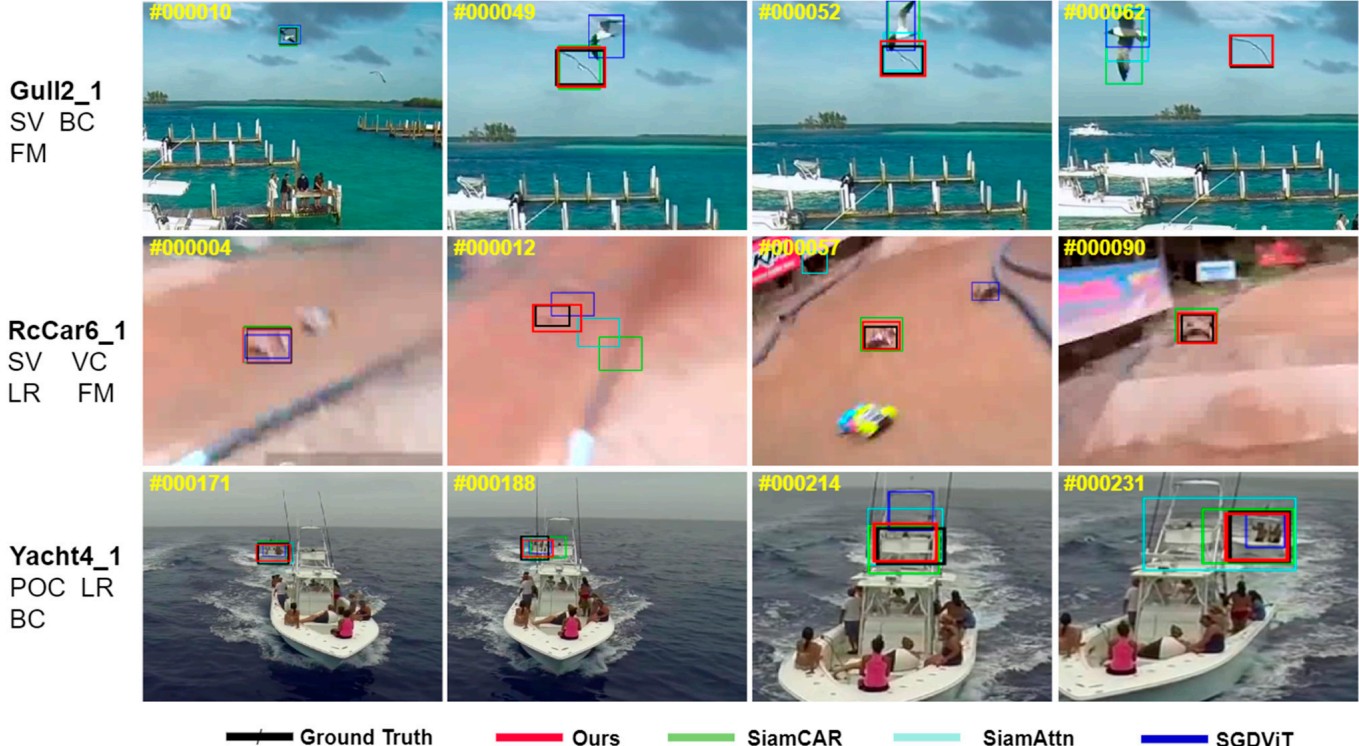

**Figure 12.** Qualitative analysis of our network and three SOTA trackers on Gull2_1, RcCar6_1 and Yacht4_1 aerial remote sensing sequences in DTB70 benchmark.

As shown in Figures 11 and 12, the above seven aerial remote sensing tracking sequences prove that the tracker proposed in this paper reveals outstanding aerial tracking performance when meeting scale variation, viewpoint change, occlusion, similar object clutter, background clutter and so on.

## 5. Conclusions and Further Work

In this work, a new effective visual tracker for aerial remote sensing tracking is proposed. This tracker combines multi-scale features with global information and prediction head attention module to address challenging scenes, such as scale variation, viewpoint change, occlusion and background clutter, and achieves accurate and robust aerial tracking. The multi-scale feature can be obtained from the feature map's channel splitting and depth-wise group convolution, and the global information is obtained by transformer-based multi-layer feature interaction. After that, the prediction head attention module can suppress the challenges of similar object and background clutter by adaptively adjusting the spatial position and channel contribution of the response map. Experiments on both aerial and general testing benchmarks demonstrate that our tracker can effectively meet the challenges of aerial tracking, improve aerial remote sensing tracking performance and achieves real-time tracking (40.8 fps).

In this work, we mainly address the challenges of scale variation, background interference and occlusion in aerial tracking to improve aerial tracking accuracy. Although the proposed algorithm can realize real-time tracking, the tracking speed is slightly decreased after adding the proposed TMGFE and PHAM. And, as the tracking speed is also a very important issue in aerial tracking, in the next work, we hope to balance the tracking speed while improving the tracking accuracy of aerial tracking to realize faster and more accurate aerial tracking.

**Author Contributions:** Conceptualization, Q.C.; methodology, Q.C.; validation, Q.C. and J.L.; investigation, Q.C., X.W. and J.L.; resources, Y.Z. and C.L.; writing—original draft preparation, Q.C.; writing review and editing, J.L. All authors have read and agreed to the published version of the manuscript.

**Funding:** This work was supported by the National Natural Science Foundation of China under Grant 61905240.

**Institutional Review Board Statement:** Not applicable.

**Informed Consent Statement:** Not applicable.

**Data Availability Statement:** Not applicable.

**Acknowledgments:** The authors are grateful for the anonymous reviewers' critical comments and constructive suggestions.

**Conflicts of Interest:** The authors declare no conflict of interest.

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
