# Peer review of "Global Multi-Scale Optimization and Prediction Head Attentional Siamese Network for Aerial Tracking"

_symmetry, doi:10.3390/sym15091629_

Round 1
Reviewer 1 Report
Report on the paper titled “Global multi-scale optimization and
prediction head attentional Siamese network for aerial
tracking”
Manuscript ID: symmetry-2549803
In this article, author try to investigate global multi-scale optimization and prediction head
attentional Siamese network for aerial tracking. Further they conduct ablation experiments on
aerial tracking benchmarks, including UAV123, UAV20L, UAV123@10fps and DTB70, to verify
the effectiveness of the proposed network. The comparisons of our tracker with several state-of-
the-art (SOTA) trackers are also conducted on four benchmarks to verify the superior performance.
The paper offers interesting insights; however, there are a few issues that the authors need
to address.
ï‚· Abstract of this article is unclear, like in the title prediction head attentional Siamese are
used, in the abstract section is unclear it is either authors defined this concept or already
exists. Please write the abstract in clear form, and explain what you want from this article.
ï‚· Please add the limitation and delimitation of your proposed model before applying it.
ï‚· Please add the detail about your proposed model, and why this is much more important
than existing models.
ï‚· Please add the physical meaning of equation (1) in detail.
ï‚· Please add the full stop at the end of each equation.
ï‚· Please explain the domain and codomain of the map defined in equation (14).
ï‚· In equation (16), in the subscript of the summation used (?, ?), not belong to any
environments. In this form this is meaningless, please add it is taken from what
environments.
ï‚· Equation (17) has no “=” sign and explain what is mean ?, ?, ?, ? > 0. I think there are
something missing, please cheek it again carefully.
ï‚· Please add further future work at the end.
ï‚· Please improve the reference list and add some more up to date reference related to your
work.

Minor editing of English language required
Reviewer 2 Report
Authors presented a siamese network for tracking objects. Two new modules were introduced, TMGFE for multi-scale feature representation and global information interaction and another module, PHAM that adds context information for more precise tracking.
Paper is well-written with clear explanations of modules. Several tests were done and very good results were achieved.
References and introduction is relevant and informative.
Perhaps only suggestion is to be more careful with beggining of sentences (in some sections there are many bigginings with “This…”).
Reviewer 3 Report
This paper has a clear explanation for the motivation of the research. The idea is innovative. And the experiment is done comprehensively. The result is sound enough to support the idea. But the figures like figure 7, 9, 10 are not clear enough. The result of the research is not presented obviously. The curves are too crowded to distinguish.
